# Effect of Al-5Ti-0.25C-0.25B and Al-5Ti-1B Master Alloys on the Microstructure and Mechanical Properties of Al-9.5Si-1.5Cu-0.8Mn-0.6Mg Alloy

**DOI:** 10.3390/ma16031246

**Published:** 2023-02-01

**Authors:** Yihan Wen, Yuying Wu, Yongjie Wu, Tong Gao, Zuoshan Wei, Xiangfa Liu

**Affiliations:** 1Key Laboratory of Liquid-Solid Structural Evolution and Processing of Materials, Ministry of Education, Shandong University, Jinan 250061, China; 2Shandong Key Laboratory of Advanced Aluminum Materials and Technology, Binzhou Institute of Technology, Binzhou 256600, China

**Keywords:** Al-Si alloy, grain refinement, mechanical properties, master alloys, extrusion

## Abstract

The goal of this research was to determine how the master alloys Al-5Ti-0.25C-0.25B and Al-5Ti-1B affected the mechanical properties and structural characteristics of the alloy Al-9.5Si-1.5Cu-0.8Mn-0.6Mg. Field emission scanning electron microscopy (FE-SEM) was used to probe the microscopic composition, and the mechanical properties were evaluated using tensile testing. The results showed that, by adding 0.5% Al-5Ti-0.25C-0.25B master alloy and 0.5% Al-5Ti-1B master alloy, the α-Al dendrites can be significantly refined. In the extrusion state, the ultimate tensile strength and elongation with 0.5% Al-5Ti-0.25C-0.25B master alloy reached 380 MPa and 11.2%, which were 5.5% and 22.4% higher than no refinement, respectively. The elongation of the samples with the Al-5Ti-1B alloy addition increased from 9% to 11.9%, which is attributed to the fact that more pronounced complete recrystallization occurred during the extrusion heat treatment.

## 1. Introduction

The Al-Si alloy has excellent casting, processing, and forming qualities, as well as strong strength and a low coefficient of thermal expansion. Amongst the cast aluminum alloys, in a variety of situations, Al-Si alloys find widespread usages, such as the production of automotive engine pistons, cylinder blocks, car chassis, wheels, and other components [1,2,3,4,5,6]. The Al-9.5Si-5Cu-0.8Mn-0.6Mg alloy is obtained by melting and casting before extrusion, during which time problems such as the segregation of alloy elements, developed dendrites, and coarse brittleness often occur, giving rise to a decrease in the alloy’s mechanical properties [7]. According to the grain refinement view, the Sr modification of eutectic silicon can optimize the alloy properties and improve its machinability and serviceability. Grain refinement methods include physical refinement, thermodynamic refinement, and chemical refinement [8,9,10]. The chemical refining methods commonly use the Al-Ti-B and Al-Ti-C master alloys. In addition to the traditional Al_3_Ti as the nucleation mass, the introduction of a small amount of C and B elements can effectively refine the grain. However, both master alloys are prone to unstable refinement effects, and the TiC and TiB_2_ particles tend to precipitation and intercalation, which reduces the quality of the alloy product. Therefore, developing the efficient grain refiners within an acceptable and manageable range is critical [11,12,13].

The Al-Si-Cu series of high tenacity cast aluminum alloys include the Aural-2, Aural-3, and Aural-5 series of aluminum alloys developed by Alcoa Canada and the AA362, AA367, and AA365 aluminum alloys developed by Mercury Marine in the USA. After heat treatment, these alloys have high tensile and yield strength, and almost all have elongation rates greater than 10%. In recent years, Rheinfelden of Germany has used multi-component micro-alloying to improve, based on the Silafont-36 alloy, and developed the Silafont-38 alloy, which has an ultimate tensile strength of 300~350 MPa, a yield strength of 230~280 MPa, and elongation of 8~11%. It is considered to be the best automotive structural part at present for casting the aluminum alloy. Some Japanese scholars found that when the Si content was controlled between 9 and 10.5 wt.% in Al-Si-Cu alloys and 0.3 to 0.5 wt.% Mg was introduced for strengthening, the castings obtained had the best overall mechanical properties [14,15,16]. However, the microstructures, as well as the mechanical characteristics of the Al-Si-Cu alloy, have yet to be increased.

In this paper, the Al-9.5Si-1.5Cu-0.8Mn-0.6Mg alloy was investigated, and different contents of the Al-5Ti-0.25C-0.25B and Al-5Ti-1B master alloy were introduced into the alloy to compare the refinement effect on the α-Al grains. The heat treatment process was then optimized to analyze the interrelationship between the variation in the organization and the properties of as-extruded.

## 2. Materials and Methods

The primary ingredients of the test samples are: industrial pure aluminum (99.7%, all ingredients are expressed in wt.% unless indicated separately), high-purity aluminum ingot (99.99%), high-purity silicon (99.9%), industrial pure magnesium (99.7%), high-purity copper (99.9%), high-purity manganese (99.9%), sponge zirconium (99.5%), titanium sponge (99.5%), and Al-10Sr alloy, auxiliary materials such as C_2_Cl_6_ refining agent and slag remover. The master alloys were the Al-5Ti-0.25C-0.25B and Al-5Ti-1B (delivered by Shandong Al and Mg Melt Technology Co. Ltd.). The composition of the Al-9.5Si-1.5Cu-0.8Mn-0.6Mg alloy is shown in Table 1.

The raw materials were proportioned according to the test composition and dissolved in a well-style crucible resistance furnace. Then, other required test raw materials were added into the Al melts. When all raw materials were completely melted, stirring began of the melting using a graphite rod. During the melting process, the temperature of the melt was monitored using a portable NiCr thermocouple, at about 750 °C. Owing to the relatively high Si content of the alloy, 0.2% Al-10Sr was added for the modification treatment. When the melt temperature settled to around 730 °C, the melt was added with 0.5%, 1.0%, and 1.5% of the two master alloys, respectively, and left to hold for 20 min, then cast into the preheated mold with the casting temperature controlled at about 720 °C, thus obtaining samples A1, A2, and A3, and B1, B2, and B3, respectively. The sample without refinement was A0. The mold is made of cast iron, forming into Φ 90 mm ∗ 300 mm cylindrical ingots [17]. The obtained ingots were homogenized at 500 °C for 8 h and subsequently extruded for the ingots of alloys A0, A1, B1 with the extrusion temperature selected as 430 °C and the extrusion ratio of 25:1 to obtain samples C0, C1, and C2. The codes, master alloy addition levels, and status of the Al-9.5Si-1.5Cu-0.8Mn-0.6Mg alloys are given in Table 2.

In accordance with Chinese National Standard GB/T228.1-2015, the test alloy was machined into a “dog bone” type. Figure 1 displays the dimensional characteristics of the tensile specimen at the standard ambient temp. After T6 heat treatment, the as-cast material’s mechanical properties were analyzed. The experiment was using the universal material testing machine (WDW-100D), which has a tension velocity of 2 mm/min. The test will be mounted on the two ends of the test bar in the upper and lower chuck of the universal testing machine and equipped with a room-temperature stretching electronic extensometer. The section cut from the tensile bar can be used for metallographic microstructure analysis and microstructural observations were carried out by mechanical polishing, where the polishing solution was MgO [18]. The corrosion solution for the alloy was the Keller solution (95% H_2_O, 1.0% HF, 1.5% HCI, 2.5% HNO_3_) [17]. An optical microscope (LMCG, Wetzlar, Germany) and a field emission scanning electron microscope (FESEM, SU-70, Hitachi, Tokyo, Japan) with an energy dispersive spectroscopy (EDS, EX-250, Horiba, Kyoto, Japan) detector at 15 kV was used to examine and detect microstructure.

## 3. Results and Discussion

### 3.1. Microstructure and Microanalysis of the Al-5Ti-1B Master Alloy and Al-5Ti-0.25C-0.25B Master Alloy

Figure 2 shows the XRD results for the two master alloys. This shows that the Al-5Ti-1B master alloy has mainly α-Al, Al_3_Ti, and TiB_2_ phases. Combined with the elemental mapping of an area in Figure 3, the Al-5Ti-0.25C-0.25B master alloy contains not only the three phases mentioned above, but also the TiC particles. The SEM image in Figure 4a shows that the TiC and TiB_2_ particles are more uniformly distributed across the matrix, with particle sizes ranging from 0.5 to 2 μm. There are two main morphologies, one appears as a slatted shape, combined with the EDS analysis of point 1, as TiB_2_ particles with a small amount of C enrichment, and the other is irregularly granular, the EDS results show it as the enriched TiC particles with the presence of B elements and a slight tendency to aggregate. Figure 4b SEM photograph shows that the matrix is mainly distributed with the massive Al_3_Ti, as well as the TiB_2_ particles, with the general size of Al_3_Ti being around 50 μm and the TiB_2_ particles being more numerous and smaller in size, almost all being below 1 μm [19].

### 3.2. Analysis of Microstructure and Mechanical Properties of As-Cast Al-9.5Si-1.5Cu-0.8Mn-0.6Mg Alloy

The Al-9.5Si-1.5Cu-0.8Mn-0.6Mg alloy was refined by different contents of the master alloy, and its microstructure is shown in Figure 5. Combined with Table 3, where the Al-dendrite size was measured by Image J software, this shows that incorporating 0.5% Al-5Ti-0.25C-0.25B master alloy caused a remarkable grain refinement, with the Al-dendrite size of 29.46 μm. When added to 1.5%, the alloy tends to coarsen compared to the former. From Figure 5e,f, the grain size decreases with the addition of 0.5% Al-5Ti-1B master alloy compared to the original alloy. While the addition further increased to 1% and 1.5%, the test alloy showed coarse columnar crystals and the α-Al grain size also became larger and the refinement effect became worse. This is due to the presence of a certain amount of Si in the Al-9.5Si-1.5Cu-0.8Mn-0.6Mg alloy. With the three additions, the refinement effect of the two master alloys was optimal at an addition level of 0.5%, while further increases in the introduced amount led to a decrease in the refinement effect, the refinement effect exerted by the Al-5Ti-0.25C-0.25B master alloy was relatively stable and the grains of the test alloy had a significant refinement effect with increasing additions, although they also had a tendency to become larger. Therefore, the Al-5Ti-0.25C-0.25B master alloy was able to perform better grain refinement compared to the Al-5Ti-1B master alloy.

The tensile tests were undertaken after the heat treatment. The effect of the refinement varies with the amount of the master alloy added, and according to fine grain strengthening, as the grain size becomes finer, the strength will increase. From Figure 6, without refinement, the ultimate tensile strength (UTS) of the original alloy was 259 MPa, the yield strength (YS) was 230 MPa, and the elongation (EI) was 1.56%. The A1 alloy has the best mechanical properties with UTS, YS, and EI of 301 MPa, 265 MPa, and 2.16%, respectively, and an increase of approximately 20%, 26%, and 54.3% in the UTS, YS, and EI, respectively, compared to A0. With the addition of the Al-5Ti-0.25C-0.25B master alloy content, the mechanical properties of alloys A2 and A3 decreased, but, overall, the properties were all improved compared to the A0. For the Al-5Ti-1B master alloy, the alloy’s mechanical characteristics also tended to decrease with the addition of the master alloy. The UTS, YS, and EI of alloy B1 were 276 MPa, 244 MPa, and 1.76%, respectively. The B3 alloy had UTS, YS, and EI of 251 MPa, 215 Mpa, and 1.4%, respectively, and the tensile characteristics are slightly reduced in comparison with the A0 alloy, suggesting that the master alloy Al-5Ti-1B lost its fineness effect at this additional level and cannot play the role of fine grain strengthening.

Figure 7 showed the fracture surface by using SEM. Figure 7d–f showed the magnified shapes at the yellow dashed boxes of a–c, whereas the red solid box in Figure 7e was the magnified photos at the higher magnification. Figure 7a showed that the tensile fracture of the original alloy without refinement was relatively flat. The fracture morphology is mainly a smooth unraveling surface, almost without dimples and a few tearing edges of short length. It exhibits the typical brittle fracture characteristics. Following the introduction of the 0.5% Al-5Ti-0.25C-0.25B master alloy, the number and length of tearing edges increased, the size of the cleavage plane decreased, and a multitude of small dimples appeared. Adding 0.5% of the Al-5Ti-1B master alloy has more cleavage planes and a slightly larger size, but it was smaller than the original alloy. There were no obvious dimples in the fracture.

After the addition of two master alloys refinement, the quantity of grains of the Al-9.5Si-1.5Cu-0.8Mn-0.6Mg alloy increased. Therefore, during the stretching process, the deformation generated can be carried out in more grains and the uniformity of the deformation was improved. In addition, the increase of the grain boundaries slowed down the crack expansion, so the strength and elongation of the specimens A1 and B1 were improved compared with those of the specimen A0. The main manifestation of the fracture was an apparent reduction in the size of the cleavage planes. Apart from this, the specimen A1 has some dimples at the fracture.

### 3.3. Heat Treatment Process Optimization

The heat treatment process is a relatively not complicated and effective way to refine the material’s performance. The heat treatment process consists of two phases: solution treatment and aging treatment. For the purpose of determining the main parameters of the heat treatment process of the tested alloy and to prepare for the alloy performance test later, the heat treatment of the as-extruded Al-9.5Si-1.5Cu-0.8Mn-0.6Mg alloy was investigated. As can be seen in the solid solution-hardness variation graph shown in Figure 8a, the as-extruded Al-9.5Si-1.5Cu-0.8Mn-0.6Mg alloy has the highest Brinell hardness in the solid solution at 525 °C for 5 h. This value was determined to be its optimum solid solution parameter. At this parameter, the alloy was treated with artificial aging. From Figure 8b, the hardness of the Al-9.5Si-1.5Cu-0.8Mn-0.6Mg alloy showed a rising and then decreasing trend with the rising aging time, the highest value of the aging toughness appeared at the 190 °C peak aging position. It was established that holding at 190 °C for 10 h was the optimal aging process parameter. Integrating the results of the two stages and considering the error of the test equipment, the optimum heat treatment process parameters of the extruded Al-9.5Si-1.5Cu-0.8Mn-0.6Mg alloy were determined to be the solution treatment at 525 ± 5 °C for 5 h and the aging treatment at 190 ± 5 °C for 10 h.

### 3.4. Analysis of Grain Microstructure and Mechanical Properties of As-Extruded Al-9.5Si-1.5Cu-0.8Mn-0.6Mg Alloy

The samples obtained from alloys A0, A1, and B1 after the extrusion process were noted as C0, C1, and C2. Figure 9 showed the EBSD analysis of the three alloys in the extruded state. From Figure 9a–c, there were plenty of small grains at the grain boundary of the three alloys. The C1 alloy had a smaller grain size compared with the C0 alloy and C2 alloy, and, at the same time, the particle distribution in the matrix was significantly improved. Figure 9d–f showed the grain size statistics of the C0, C1, and C2 alloys. The unrefined Al-9.5Si-1.5Cu-0.8Mn-0.6Mg alloy has a relatively coarse grain pattern of grain size of approximately 35 μm. The average grain sizes of the C1 and C2 alloys improved by adding the two master alloys were about 31 μm and 33 μm, which were reduced by 11.4% and 5.7%, respectively. The C1 alloy has the smallest grain size, further explaining its refinement effect over the Al-5Ti-1B master alloy.

The room temperature tensile properties of the alloys C0, C1, and C2 were tested in the extruded state under the optimum heat treatment process, which is the solution approach at 525 ± 5 °C for 5 h and the aging treatment at 190 ± 5 °C for 10 h. From Figure 10b, it can be seen that the UTS, YS, and EI of the C0 alloy in the extruded state without refinement were 360 MPa, 307 MPa, and 9%, respectively. The UTS, YS, and EI are improved after extrusion; when the content of the Al-5Ti-1B master alloy is 0.5%, they reached 380 MPa, 330 MPa, and 11.2%, respectively. Compared to the situation without refinement, the EI of the Al-9.5Si-1.5Cu-0.8Mn-0.6Mg alloy after adding 0.5% Al-5Ti-1B master alloy increased by about 32.2% at room temperature.

The fracture of the Al-9.5Si-1.5Cu-0.8Mn-0.6Mg alloy after extrusion was depicted in Figure 11. From the fracture morphology, the fracture mechanism of the as-extruded Al-9.5Si-1.5Cu-0.8Mn-0.6Mg alloy was a ductile fracture. Unlike the alloy in the as-cast state, more dimples existed in the fractures of alloys C0, C1, and C2, while no obvious cleavage planes were observed. The C0 alloy has small and shallow dimples at the fracture and exhibits low plasticity in its tensile properties. After adding 0.5% Al-5Ti-0.25C-0.25B master alloy and 0.5% Al-5Ti-1B master alloy, the dimples become bigger and deeper than without refinement. Meanwhile, enhanced plasticity characterized the alloy throughout the tensile procedure. Additionally, the surface smoothness at the fracture of the C0 and C2 alloys were higher than the C1 alloy, which will cause fast intergranular crack propagation and low tensile strength. When added the master alloy, the C1 and C2 alloys had reduced grain size and an increased number of grain boundaries. The driving power of recrystallization is enhanced during the process of extrusion, so recrystallization can easily occur. Furthermore, the Zr element in the alloy inhibits the recrystallization of the alloy by forming the substable phase Al_3_Zr (L1_2_) [20,21,22]. However, the Zr element is easily reacted with the addition of Al_3_Ti, which weakens the inhibition of the recrystallization behavior of the alloy by the Zr element and also affects the TiB_2_ particles to exerting the influence of heterogeneous nucleation, reducing the refining effect. As a result, the C1 and C2 alloy plasticity was higher than the C0 alloy, and the C2 alloy elongation was the highest.

## 4. Conclusions

The influence of the Al-5Ti-0.25C-0.25B master alloy and the Al-5Ti-1B master alloy on the microstructure and properties of the Al-9.5Si-1.5Cu-0.8Mn-0.6Mg alloy were investigated.

At the same addition level, Al-5Ti-0.25C-0.25B shows improved grain refinement properties. The mechanical properties of the Al-9.5Si-1.5Cu-0.8Mn-0.6Mg alloy were significantly improved when refined with 0.5% of the Al-5Ti-0.25C-0.25B master alloy. The ultimate tensile strength, yield strength, and elongation reached 301 MPa, 265 MPa, and 2.16%, respectively. The tensile properties of the Al-9.5Si-1.5Cu-0.8Mn-0.6Mg alloy are also improved in the 0.5% Al-5Ti-1B master alloy.The α-Al’s average grain sizes, with the addition of the Al-5Ti-0.25C-0.25B master alloy and the Al-5Ti-1B master alloy, were about 31 μm and 33 μm in the extruded state, which was lower than 35 μm of the original alloy without refinement.The mechanical properties test results showed that the ultimate tensile strength, yield strength, and elongation of the as-extruded alloy were increased to 380 MPa, 330 MPa, and 11.2%, respectively, compared with no refinement when increased by 5.5%, 7.5%, and 24.4%, respectively. The elongation property can be improved considerably, including with a master alloy of 0.5% Al-5Ti-1B.

## Figures and Tables

**Figure 1 materials-16-01246-f001:**
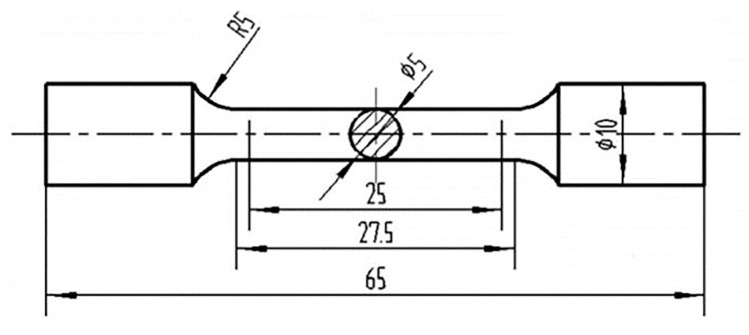
Schematic diagram of tensile rod at room temperature.

**Figure 2 materials-16-01246-f002:**
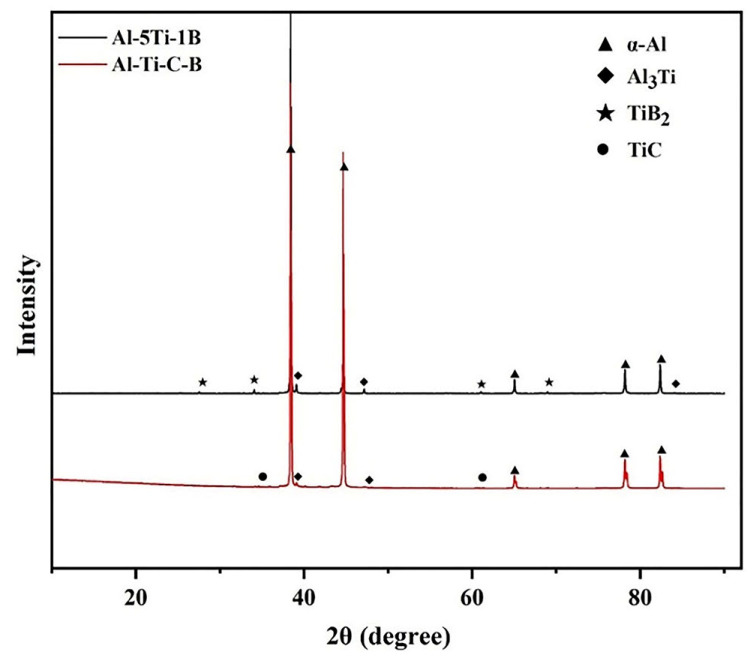
XRD patterns of Al-5Ti-1B master alloy and Al-5Ti-0.25C-0.25B master alloy.

**Figure 3 materials-16-01246-f003:**
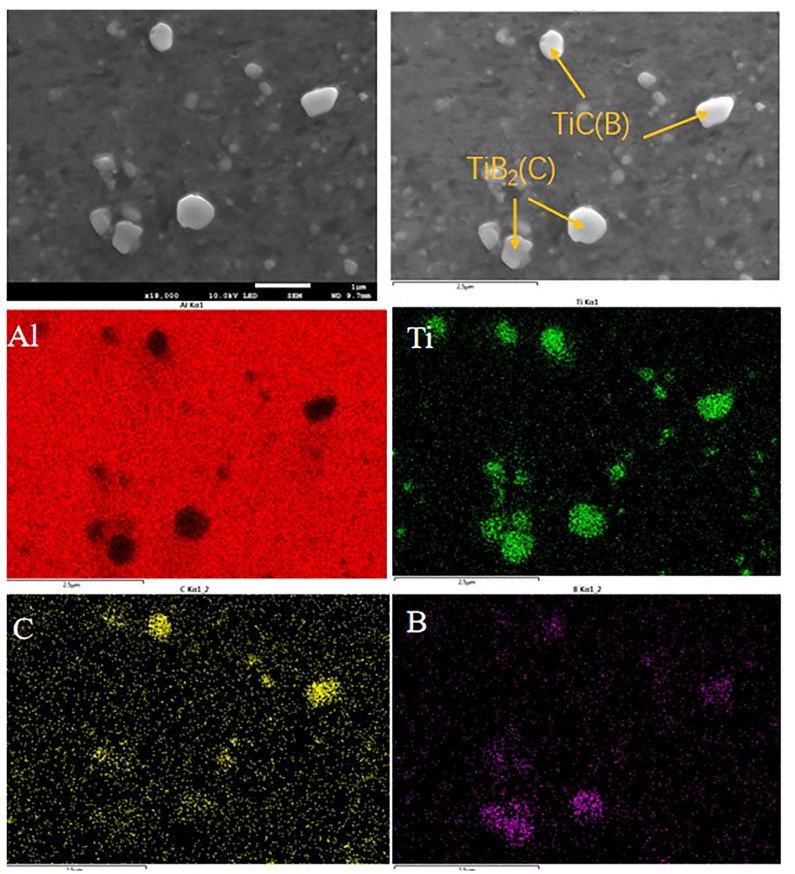
Element mapping of an area in the Al-5Ti-0.25C-0.25B alloy.

**Figure 4 materials-16-01246-f004:**
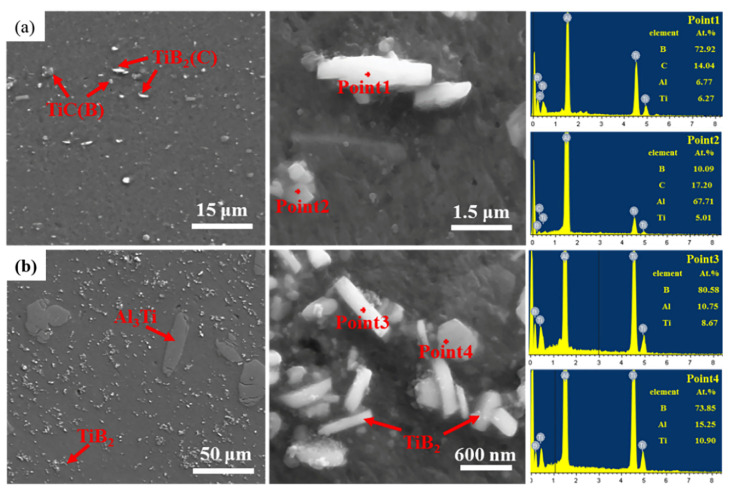
SEM analysis and EDS point analysis: (**a**) Al-5Ti-0.25C-0.25B; (**b**) Al-5Ti-1B.

**Figure 5 materials-16-01246-f005:**
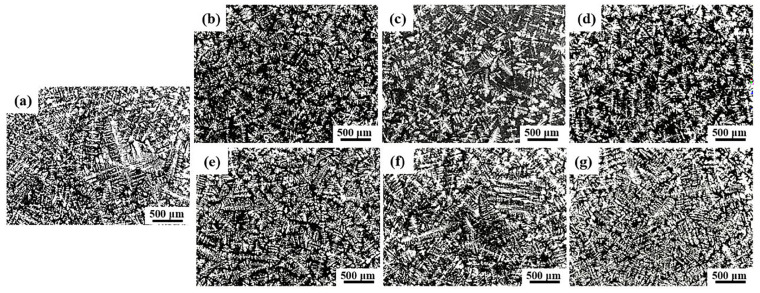
Microstructure of Al-9.5Si-1.5Cu-0.8Mn-0.6Mg alloys: (**a**) A0; (**b**) A1; (**c**) A2; (**d**) A3; (**e**) B1; (**f**) B2; (**g**) B3.

**Figure 6 materials-16-01246-f006:**
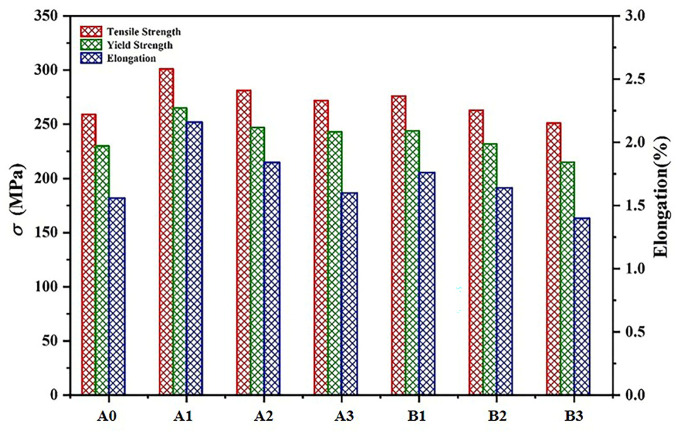
Strength and elongation of as-cast Al-9.5Si-1.5Cu-0.8Mn-0.6Mg alloy.

**Figure 7 materials-16-01246-f007:**
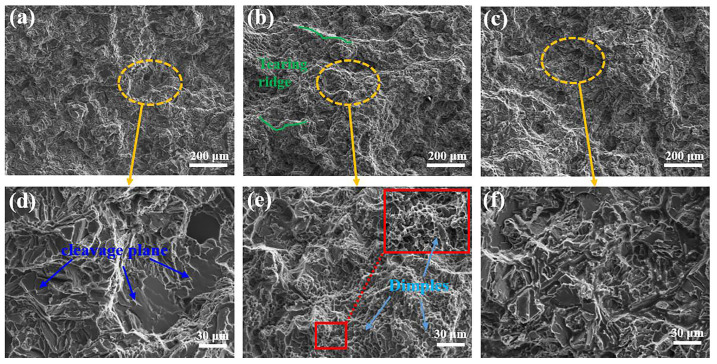
Fracture of as-cast Al-9.5Si-1.5Cu-0.8Mn-0.6Mg alloy: (**a**,**d**) A0 alloy; (**b**,**e**) A1 alloy; (**c**,**f**) B1 alloy.

**Figure 8 materials-16-01246-f008:**
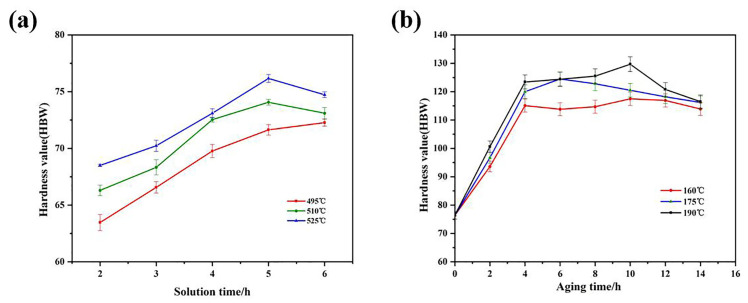
As-extruded Al-9.5Si-1.5Cu-0.8Mn-0.6Mg alloy: (**a**) solution-hardness curve; (**b**) aging-hardness curve.

**Figure 9 materials-16-01246-f009:**
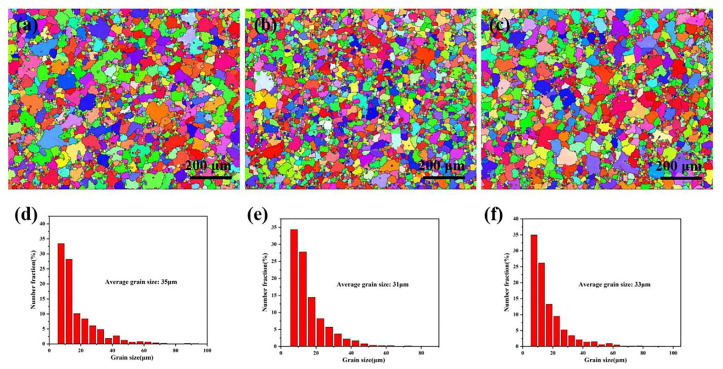
EBSD analysis and average grain size statistics of as-extruded Al-9.5Si-1.5Cu-0.8Mn-0.6Mg alloy: (**a**,**d**) C0 alloy; (**b**,**e**) C1 alloy; (**c**,**f**) C2 alloy.

**Figure 10 materials-16-01246-f010:**
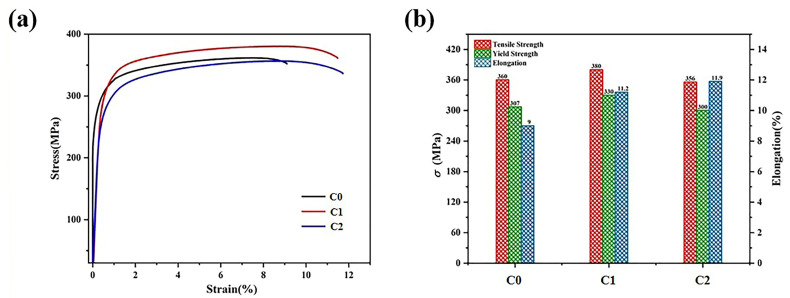
Room temperature tensile properties of as-extruded Al-9.5Si-1.5Cu-0.8Mn-0.6Mg alloy (**a**,**b**).

**Figure 11 materials-16-01246-f011:**
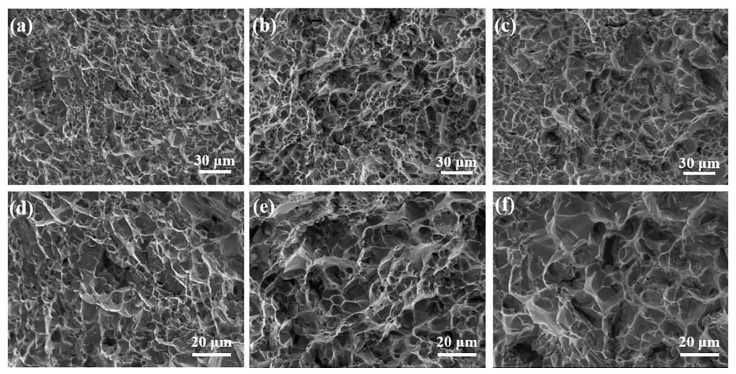
Fracture morphology of the as-extruded Al-9.5Si-1.5Cu-0.8Mn-0.6Mg alloy: (**a**,**d**) C0 alloy; (**b**,**e**) C1 alloy; (**c**,**f**) C2 alloy.

**Table 1 materials-16-01246-t001:** Chemical composition of Al-9.5Si-1.5Cu-0.8Mn-0.6Mg alloy (wt.%).

Elements	Si	Cu	Mn	Mg	Fe	Zr	Ti	Al
Nominal	9.50	1.50	0.80	0.60	0.10	0.10	0.06	Bal.

**Table 2 materials-16-01246-t002:** Summary of the text alloys with different master alloy addition.

Sample	Addition Content (wt.%)	Master alloy	State
A0	-	-	As-cast
A1	0.5	Al-5Ti-0.25C-0.25B	As-cast
A2	1.0	Al-5Ti-0.25C-0.25B	As-cast
A3	1.5	Al-5Ti-0.25C-0.25B	As-cast
B1	0.5	Al-5Ti-1B	As-cast
B2	1.0	Al-5Ti-1B	As-cast
B3	1.5	Al-5Ti-1B	As-cast
C0	-	-	As-extruded
C1	0.5	Al-5Ti-0.25C-0.25B	As-extruded
C2	0.5	Al-5Ti-1B	As-extruded

**Table 3 materials-16-01246-t003:** The Al-dendrite size of the SDAS of the alloy after refinement with Al-5Ti-0.25C-0.25B and Al-5Ti-1B master alloy.

Sample	A0	A1	A2	A3	B1	B2	B3
SDAS (μm)	53.40	29.46	39.55	41.55	35.36	42.72	54.31

## Data Availability

The data presented in this study are available in article.

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
