# Peer review of "Effect of Al-5Ti-0.25C-0.25B and Al-5Ti-1B Master Alloys on the Microstructure and Mechanical Properties of Al-9.5Si-1.5Cu-0.8Mn-0.6Mg Alloy"

_materials, 2023, doi:10.3390/ma16031246_

Round 1

Reviewer 1 Report

This manuscript focuses on the effect of the Al-5Ti-0.25C-0.25B master alloy and the Al-5Ti-1B master alloy on the microstructure and mechanical properties of the Al-9.5Si-1.5Cu-0.8Mn-0.6Mg alloy. This is indeed a more demanding subject, and Al-Si based alloys have implications for casting aluminum alloys and automotive industry production. By adding small fractions of 0.5–1.0% of the master alloy Al-5Ti-0.25C-0.25B to the Al-9.5Si-1.5Cu-0.8Mn-0.6Mg alloy could significantly improve the mechanical properties of the alloy. The overall design, methodology, and results are reasonable and would be of great importance to the readership of Materials Journal. However, before accepting this manuscript, a few questions need to be addressed and incorporated into the manuscript.

  1. The identification of the TiB2, TiC, Al3Ti phases from the EDX point spectra did not prove the correct stoichiometry of these phases. For instance, check the point spectra 1 of TiB2 phase of the Al-5Ti-0.25C-0.25B master alloy, which yielded 72 at. % B and only 6-7 at at. % Ti and Al, while carbon fractions were approx. 15 at.%. Also, authors pointed out that they have trace amount of carbon, even carbon composition was significantly higher than the Ti. Any explanation for this?
  1. Why did you choose 15 kV for the EDX? How can you ensure that there are no other characteristic x-rays contributing to these quantifications? Why not showing the chemical mapping in SEM or, preferable, TEM to prove the phase composition? New chemical maps or spectra will be acquired and should be added in the manuscript.
  1. Page 4, Section 3.2, Paragraph 1, line 132, the authors pointed out that "At the same addition amount, the Al-9.5Si-1.5Cu-0.8Mn-0.6Mg alloy has finer grains with the Al-5Ti-0.25C-0.25B master alloy, which indicates that the Al-5Ti-0.25C-0.25B master alloy has resistance to Si "poisoning." As a result, the Al-5Ti-0.25C-0.25B master alloy has a much better grain refining performance than the Al-5Ti-1B master alloy. Do you have any evidence that proves that the Al-5Ti-0.25C-0.25B master alloy has resistance to Si "poisoning"?
  1. From Fig. 3, with a coarse scale bar of 500 um, it is even more difficult to prove that the grain refinement happened after adding the small amount of master alloys. To confirm the refinement, the manuscript should also include high-magnification SE images.
  1. Page 4, para 2, the authors mention that "after adding 0.5% Al-2Ti-0.5B-0.5C master alloy, the UTS, YS, and EI of the alloy were 301 MPa, 265 MPa, and 2.16%, respectively." Is this a typo? or did you use the Al-2Ti-0.5B-0.5C master alloy. From the manuscript, it is evident that you have used only Al-5Ti-0.5B-0.5C and Al-5Ti-1B master alloys.
  1. Figure 4. Strength and elongation of as-cast Al-9.5Si-1.5Cu-0.8Mn-0.6Mg alloy is somehow misleading with wrong annotations on the x-axis. Why did you used 1,2,3,4,5,6 , why not using A0, A1, A2…… This should be checked and fixed thoroughly.

Reviewer 2 Report

Notes :

1- In the abstract, there is repetition in the names of the alloys, as each alloy used in the research was mentioned three times, which requires a linguistic reformulation of the abstract.

2- In paragraph 2- Materials and Methods, no reference was made to the pouring temperature of the castings in the mold, as well as the material and dimensions of the mold.

3- The standard sample for tensile testing was not referred to within any standard.

4- Paragraph 2 of the paper did not include a reference to preparing samples for microstructure testing, for example, referring to an etching solution for this type of alloy.

5- From the results of the mechanical properties, Figure No. (4), there is no clear difference in the values of the mechanical properties for samples 2, 3, 4, 5, and there is a convergence in the values for samples 0 and 6. These results need clarification and strengthening discussion.

6- It was preferable to measure the SDAS of Al-dendrite to verify its refining as mentioned in the paper text.

Round 2

Reviewer 1 Report

With the addition of new data, It is evident that the overall quality of the manuscript has been improved. The author’s addressed most of mu questions with great satisfaction. This manuscript is now ready for publication in Materials.